# Scalable Log Determinants for Gaussian Process Kernel Learning

**Kun Dong** [1], **David Eriksson** [1], **Hannes Nickisch** [2], **David Bindel** [1], **Andrew Gordon Wilson** [1]

[1] Cornell University, [2] Phillips Research Hamburg

## Abstract

For applications as varied as Bayesian neural networks, determinantal point processes, elliptical graphical models, and kernel learning for Gaussian processes (GPs), one must compute a log determinant of an $n \times n$ positive definite matrix, and its derivatives – leading to prohibitive $\mathcal{O}(n^3)$ computations. We propose novel $\mathcal{O}(n)$ approaches to estimating these quantities from only fast matrix vector multiplications (MVMs). These stochastic approximations are based on Chebyshev, Lanczos, and surrogate models, and converge quickly even for kernel matrices that have challenging spectra. We leverage these approximations to develop a scalable Gaussian process approach to kernel learning. We find that Lanczos is generally superior to Chebyshev for kernel learning, and that a surrogate approach can be highly efficient and accurate with popular kernels.

## 1 Introduction

There is a pressing need for scalable machine learning approaches to extract rich statistical structure from large datasets. A common bottleneck — arising in determinantal point processes [1], Bayesian neural networks [2], model comparison [3], graphical models [4], and Gaussian process kernel learning [5] — is computing a log determinant over a large positive definite matrix. While we can approximate log determinants by existing stochastic expansions relying on matrix vector multiplications (MVMs), these approaches make assumptions, such as near-uniform eigenspectra [6], which are unsuitable in machine learning contexts. For example, the popular RBF kernel gives rise to rapidly decaying eigenvalues. Moreover, while standard approaches, such as stochastic power series, have reasonable asymptotic complexity in the rank of the matrix, they require too many terms (MVMs) for the precision necessary in machine learning applications.

Gaussian processes (GPs) provide a principled probabilistic kernel learning framework, for which a log determinant is of foundational importance. Specifically, the *marginal likelihood* of a Gaussian process is the probability of data given only kernel hyper-parameters. This utility function for kernel learning compartmentalizes into automatically calibrated model fit and complexity terms — called *automatic Occam's razor* — such that the simplest models which explain the data are automatically favoured [7, 5], without the need for approaches such as cross-validation, or regularization, which can be costly, heuristic, and involve substantial hand-tuning and human intervention. The automatic complexity penalty, called the *Occam's factor* [3], is a log determinant of a kernel (covariance) matrix, related to the volume of solutions that can be expressed by the Gaussian process.

Many current approaches to scalable Gaussian processes [e.g., 8–10] focus on inference assuming a fixed kernel, or use approximations that do not allow for very flexible kernel learning [11], due to poor scaling with number of basis functions or inducing points. Alternatively, approaches which exploit algebraic structure in kernel matrices can provide highly expressive kernel learning [12], but are essentially limited to grid structured data.

Recently, Wilson and Nickisch [13] proposed the *structured kernel interpolation* (SKI) framework, which generalizes structuring exploiting methods to arbitrarily located data. SKI works by providing accurate and fast matrix vector multiplies (MVMs) with kernel matrices, which can then be used in iterative solvers such as linear conjugate gradients for scalable GP inference. However, evaluating the marginal likelihood and its derivatives, for kernel learning, has followed a scaled eigenvalue approach [12, 13] instead of iterative MVM approaches. This approach can be inaccurate, and relies on a fast eigendecomposition of a structured matrix, which is not available in many consequential situations where fast MVMs are available, including: (i) additive covariance functions, (ii) multi-task learning, (iii) change-points [14], and (iv) diagonal corrections to kernel approximations [15]. Fiedler [16] and Weyl [17] bounds have been used to extend the scaled eigenvalue approach [18, 14], but are similarly limited. These extensions are often very approximate, and do not apply beyond sums of two and three matrices, where each matrix in the sum must have a fast eigendecomposition.

In machine learning there has recently been renewed interest in MVM based approaches to approximating log determinants, such as the Chebyshev [19] and Lanczos [20] based methods, although these approaches go back at least two decades in quantum chemistry computations [21]. Independently, several authors have proposed various methods to compute derivatives of log determinants [22, 23]. But *both* the log determinant *and* the derivatives are needed for efficient GP marginal likelihood learning: the derivatives are required for gradient-based optimization, while the log determinant itself is needed for model comparison, comparisons between the likelihoods at local maximizers, and fast and effective choices of starting points and step sizes in a gradient-based optimization algorithm.

In this paper, we develop novel scalable and general purpose Chebyshev, Lanczos, and surrogate approaches for efficiently and accurately computing both the log determinant and its derivatives simultaneously. Our methods use only fast MVMs, and re-use the same MVMs for both computations. In particular:

- We derive fast methods for simultaneously computing the log determinant and its derivatives by stochastic Chebyshev, stochastic Lanczos, and surrogate models, from MVMs alone. We also perform an error analysis and extend these approaches to higher order derivatives.

- These methods enable fast GP kernel learning whenever fast MVMs are possible, including applications where alternatives such as scaled eigenvalue methods (which rely on fast eigendecompositions) are not, such as for (i) diagonal corrections for better kernel approximations, (ii) additive covariances, (iii) multi-task approaches, and (iv) non-Gaussian likelihoods.

- We illustrate the performance of our approach on several large, multi-dimensional datasets, including a consequential crime prediction problem, and a precipitation problem with $n = 528,474$ training points. We consider a variety of kernels, including deep kernels [24], diagonal corrections, and both Gaussian and non-Gaussian likelihoods.

- We have released code and tutorials as an extension to the GPML library [25] at `https://github.com/kd383/GPML_SLD`. A Python implementation of our approach is also available through the GPyTorch library: `https://github.com/jrg365/gpytorch`.

When using our approach in conjunction with SKI [13] for fast MVMs, GP kernel learning is $\mathcal{O}(n + g(m))$, for $m$ inducing points and $n$ training points, where $g(m) \leq m \log m$. With algebraic approaches such as SKI we also do not need to worry about quadratic storage in inducing points, since symmetric Toeplitz and Kronecker matrices can be stored with at most linear cost, without needing to explicitly construct a matrix.

Although we here use SKI for fast MVMs, we emphasize that the proposed iterative approaches are generally applicable, and can easily be used in conjunction with *any* method that admits fast MVMs, including classical inducing point methods [8], finite basis expansions [9], and the popular stochastic variational approaches [10]. Moreover, stochastic variational approaches can naturally be combined with SKI to further accelerate MVMs [26].

We start in §2 with an introduction to GPs and kernel approximations. In §3 we introduce stochastic trace estimation and Chebyshev (§3.1) and Lanczos (§3.2) approximations. In §4, we describe the different sources of error in our approximations. In §5 we consider experiments on several large real-world data sets. We conclude in §6. The supplementary materials also contain several additional experiments and details.

## 2 Background

A Gaussian process (GP) is a collection of random variables, any finite number of which have a joint Gaussian distribution [e.g., 5]. A GP can be used to define a distribution over functions $f(x) \sim \mathcal{GP}(\mu(x), k(x, x'))$, where each function value is a random variable indexed by $x \in \mathbb{R}^d$, and $\mu : \mathbb{R}^d \to \mathbb{R}$ and $k : \mathbb{R}^d \times \mathbb{R}^d \to \mathbb{R}$ are the mean and covariance functions of the process.

The covariance function is often chosen to be an RBF or Matérn kernel (see the supplementary material for more details). We denote any kernel hyperparameters by the vector $\theta$. To be concise we will generally not explicitly denote the dependence of $k$ and associated matrices on $\theta$.

For any locations $X = \{x_1, \ldots, x_n\} \subset \mathbb{R}^d$, $f_X \sim \mathcal{N}(\mu_X, K_{XX})$ where $f_X$ and $\mu_X$ represent the vectors of function values for $f$ and $\mu$ evaluated at each of the $x_i \in X$, and $K_{XX}$ is the matrix whose $(i, j)$ entry is $k(x_i, x_j)$. Suppose we have a vector of corresponding function values $y \in \mathbb{R}^n$, where each entry is contaminated by independent Gaussian noise with variance $\sigma^2$. Under a Gaussian process prior depending on the covariance hyperparameters $\theta$, the log marginal likelihood is given by

$$\mathcal{L}(\theta|y) = -\frac{1}{2}\left[(y - \mu_X)^T \alpha + \log|\tilde{K}_{XX}| + n\log 2\pi\right] \tag{1}$$

where $\alpha = \tilde{K}_{XX}^{-1}(y - \mu_X)$ and $\tilde{K}_{XX} = K_{XX} + \sigma^2 I$. Optimization of (1) is expensive, since the cheapest way of evaluating $\log|\tilde{K}_{XX}|$ and its derivatives without taking advantage of the structure of $\tilde{K}_{XX}$ involves computing the $\mathcal{O}(n^3)$ Cholesky factorization of $\tilde{K}_{XX}$. $\mathcal{O}(n^3)$ computations is too expensive for inference and learning beyond even just a few thousand points.

A popular approach to GP scalability is to replace the exact kernel $k(x, z)$ by an approximate kernel that admits fast computations [8]. Several methods approximate $k(x, z)$ via *inducing points* $U = \{u_j\}_{j=1}^m \subset \mathbb{R}^d$. An example is the subset of regressor (SoR) kernel:

$$k^{SoR}(x, z) = K_{xU}K_{UU}^{-1}K_{Uz}$$

which is a low-rank approximation [27]. The SoR matrix $K_{XX}^{\text{SoR}} \in \mathbb{R}^{n \times n}$ has rank at most $m$, allowing us to solve linear systems involving $\tilde{K}_{XX}^{\text{SoR}} = K_{XX}^{\text{SoR}} + \sigma^2 I$ and to compute $\log|\tilde{K}_{XX}^{\text{SoR}}|$ in $\mathcal{O}(m^2 n + m^3)$ time. Another popular kernel approximation is the fully independent training conditional (FITC), which is a diagonal correction of SoR so that the diagonal is the same as for the original kernel [15]. Thus kernel matrices from FITC have low-rank plus diagonal structure. This modification has had exceptional practical significance, leading to improved point predictions and much more realistic predictive uncertainty [8, 28], making FITC arguably the most popular approach for scalable Gaussian processes.

Wilson and Nickisch [13] provides a mechanism for fast MVMs through proposing the structured kernel interpolation (SKI) approximation,

$$K_{XX} \approx WK_{UU}W^T \tag{2}$$

where $W$ is an $n$-by-$m$ matrix of interpolation weights; the authors of [13] use local cubic interpolation so that $W$ is sparse. The sparsity in $W$ makes it possible to naturally exploit algebraic structure (such as Kronecker or Toeplitz structure) in $K_{UU}$ when the inducing points $U$ are on a grid, for extremely fast matrix vector multiplications with the approximate $K_{XX}$ even if the data inputs $X$ are arbitrarily located. For instance, if $K_{UU}$ is Toeplitz, then each MVM with the approximate $K_{XX}$ costs only $\mathcal{O}(n + m\log m)$. By contrast, placing the inducing points $U$ on a grid for classical inducing point methods, such as SoR or FITC, does not result in substantial performance gains, due to the costly cross-covariance matrices $K_{xU}$ and $K_{Uz}$.

## 3 Methods

Our goal is to estimate, for a symmetric positive definite matrix $\tilde{K}$,

$$\log|\tilde{K}| = \text{tr}(\log(\tilde{K})) \quad \text{and} \quad \frac{\partial}{\partial\theta_i}\left[\log|\tilde{K}|\right] = \text{tr}\left(\tilde{K}^{-1}\left(\frac{\partial\tilde{K}}{\partial\theta_i}\right)\right),$$

where $\log$ is the matrix logarithm [29]. We compute the traces involved in both the log determinant and its derivative via *stochastic trace estimators* [30], which approximate the trace of a matrix using only matrix vector products.

The key idea is that for a given matrix $A$ and a random probe vector $z$ with independent entries with mean zero and variance one, then $\text{tr}(A) = \mathbb{E}[z^T A z]$; a common choice is to let the entries of the probe vectors be Rademacher random variables. In practice, we estimate the trace by the sample mean over $n_z$ independent probe vectors. Often surprisingly few probe vectors suffice.

To estimate $\text{tr}(\log(\tilde{K}))$, we need to multiply $\log(\tilde{K})$ by probe vectors. We consider two ways to estimate $\log(\tilde{K})z$: by a polynomial approximation of $\log$ or by using the connection between the Gaussian quadrature rule and the Lanczos method [19, 20]. In both cases, we show how to re-use the same probe vectors for an inexpensive coupled estimator of the derivatives. In addition, we may use standard radial basis function interpolation of the log determinant evaluated at a few systematically chosen points in the hyperparameter space as an inexpensive surrogate for the log determinant.

## 3.1 Chebyshev

Chebyshev polynomials are defined by the recursion

$$T_0(x) = 1, \quad T_1(x) = x, \quad T_{j+1}(x) = 2xT_j(x) - T_{j-1}(x) \text{ for } j \geq 1.$$

For $f : [-1, 1] \to \mathbb{R}$ the Chebyshev interpolant of degree $m$ is

$$f(x) \approx p_m(x) := \sum_{j=0}^{m} c_j T_j(x), \quad \text{where } c_j = \frac{2 - \delta_{j0}}{m+1} \sum_{k=0}^{m} f(x_k) T_j(x_k)$$

where $\delta_{j0}$ is the Kronecker delta and $x_k = \cos(\pi(k+1/2)/(m+1))$ for $k = 0, 1, 2, \ldots, m$; see [31]. Using the Chebyshev interpolant of $\log(1 + \alpha x)$, we approximate $\log|\tilde{K}|$ by

$$\log|\tilde{K}| - n \log \beta = \log|I + \alpha B| \approx \sum_{j=0}^{m} c_j \, \text{tr}(T_j(B))$$

when $B = (\tilde{K}/\beta - 1)/\alpha$ has eigenvalues $\lambda_i \in (-1, 1)$.

For stochastic estimation of $\text{tr}(T_j(B))$, we only need to compute $z^T T_j(B) z$ for each given probe vector $z$. We compute vectors $w_j = T_j(B)z$ and $\partial w_j / \partial \theta_i$ via the coupled recurrences

$$w_0 = z, \quad w_1 = Bz, \quad w_{j+1} = 2Bw_j - w_{j-1} \text{ for } j \geq 1,$$
$$\frac{\partial w_0}{\partial \theta_i} = 0, \quad \frac{\partial w_1}{\partial \theta_i} = \frac{\partial B}{\partial \theta_i} z, \quad \frac{\partial w_{j+1}}{\partial \theta_i} = 2\left(\frac{\partial B}{\partial \theta_i} w_j + B \frac{\partial w_j}{\partial \theta_i}\right) - \frac{\partial w_{j-1}}{\partial \theta_i} \text{ for } j \geq 1.$$

This gives the estimators

$$\log|\tilde{K}| \approx \mathbb{E}\left[\sum_{j=0}^{m} c_j z^T w_j\right] \quad \text{and} \quad \frac{\partial}{\partial \theta_i} \log|\tilde{K}| \approx \mathbb{E}\left[\sum_{j=0}^{m} c_j z^T \frac{\partial w_j}{\partial \theta_i}\right].$$

Thus, each derivative of the approximation costs two extra MVMs per term.

## 3.2 Lanczos

We can also approximate $z^T \log(\tilde{K})z$ via a Lanczos decomposition; see [32] for discussion of a Lanczos-based computation of $z^T f(\tilde{K})z$ and [20, 21] for stochastic Lanczos estimation of log determinants. We run $m$ steps of the Lanczos algorithm, which computes the decomposition

$$\tilde{K}Q_m = Q_m T + \beta_m q_{m+1} e_m^T$$

where $Q_m = [q_1 \quad q_2 \quad \ldots q_m] \in \mathbb{R}^{n \times m}$ is a matrix with orthonormal columns such that $q_1 = z/\|z\|$, $T \in \mathbb{R}^{m \times m}$ is tridiagonal, $\beta_m$ is the residual, and $e_m$ is the $m$th Cartesian unit vector. We estimate

$$z^T f(\tilde{K})z \approx e_1^T f(\|z\|^2 T)e_1 \tag{3}$$

where $e_1$ is the first column of the identity. The Lanczos algorithm is numerically unstable. Several practical implementations resolve this issue [33, 34]. The approximation (3) corresponds to a Gauss quadrature rule for the Riemann-Stieltjes integral of the measure associated with the eigenvalue

distribution of $\tilde{K}$. It is exact when $f$ is a polynomial of degree up to $2m - 1$. This approximation is also exact when $\tilde{K}$ has at most $m$ distinct eigenvalues, which is particularly relevant to Gaussian process regression, since frequently the kernel matrices only have a small number of eigenvalues that are not close to zero.

The Lanczos decomposition also allows us to estimate derivatives of the log determinant at minimal cost. Via the Lanczos decomposition, we have

$$\hat{g} = Q_m(T^{-1}e_1\|z\|) \approx \tilde{K}^{-1}z.$$

This approximation requires no additional matrix vector multiplications beyond those used to compute the Lanczos decomposition, which we already used to estimate $\log(\tilde{K})z$; in exact arithmetic, this is equivalent to $m$ steps of CG. Computing $\hat{g}$ in this way takes $\mathcal{O}(mn)$ additional time; subsequently, we only need one matrix-vector multiply by $\partial \tilde{K}/\partial\theta_i$ for each probe vector to estimate $\mathrm{tr}(\tilde{K}^{-1}(\partial\tilde{K}/\partial\theta_i)) = \mathbb{E}[(\tilde{K}^{-1}z)^T(\partial\tilde{K}/\partial\theta_i)z]$.

### 3.3 Diagonal correction to SKI

The SKI approximation may provide a poor estimate of the diagonal entries of the original kernel matrix for kernels with limited smoothness, such as the Matérn kernel. In general, diagonal corrections to scalable kernel approximations can lead to great performance gains. Indeed, the popular FITC method [15] is exactly a diagonal correction of subset of regressors (SoR).

We thus modify the SKI approximation to add a diagonal matrix $D$,

$$K_{XX} \approx WK_{UU}W^T + D, \tag{4}$$

such that the diagonal of the approximated $K_{XX}$ is exact. In other words, $D$ substracts the diagonal of $WK_{UU}W^T$ and adds the true diagonal of $K_{XX}$. This modification is not possible for the scaled eigenvalue method for approximating log determinants in [13], since adding a diagonal matrix makes it impossible to approximate the eigenvalues of $K_{XX}$ from the eigenvalues of $K_{UU}$.

However, Eq. (4) still admits fast MVMs and thus works with our approach for estimating the log determinant and its derivatives. Computing $D$ with SKI costs only $\mathcal{O}(n)$ flops since $W$ is sparse for local cubic interpolation. We can therefore compute $(W^Te_i)^TK_{UU}(W^Te_i)$ in $\mathcal{O}(1)$ flops.

### 3.4 Estimating higher derivatives

We have already described how to use stochastic estimators to compute the log marginal likelihood and its first derivatives. The same approach applies to computing higher-order derivatives for a Newton-like iteration, to understand the sensitivity of the maximum likelihood parameters, or for similar tasks. The first derivatives of the full log marginal likelihood are

$$\frac{\partial\mathcal{L}}{\partial\theta_i} = -\frac{1}{2}\left[\mathrm{tr}\left(\tilde{K}^{-1}\frac{\partial\tilde{K}}{\partial\theta_i}\right) - \alpha^T\frac{\partial\tilde{K}}{\partial\theta_i}\alpha\right]$$

and the second derivatives of the two terms are

$$\frac{\partial^2}{\partial\theta_i\partial\theta_j}\left[\log|\tilde{K}|\right] = \mathrm{tr}\left(\tilde{K}^{-1}\frac{\partial^2\tilde{K}}{\partial\theta_i\partial\theta_j} - \tilde{K}^{-1}\frac{\partial\tilde{K}}{\partial\theta_i}\tilde{K}^{-1}\frac{\partial\tilde{K}}{\partial\theta_j}\right),$$

$$\frac{\partial^2}{\partial\theta_i\partial\theta_j}\left[(y - \mu_X)^T\alpha\right] = 2\alpha^T\frac{\partial\tilde{K}}{\partial\theta_i}\tilde{K}^{-1}\frac{\partial\tilde{K}}{\partial\theta_j}\alpha - \alpha^T\frac{\partial^2\tilde{K}}{\partial\theta_i\partial\theta_j}\alpha.$$

Superficially, evaluating the second derivatives would appear to require several additional solves above and beyond those used to estimate the first derivatives of the log determinant. In fact, we can get an unbiased estimator for the second derivatives with no additional solves, but only fast products with the derivatives of the kernel matrices. Let $z$ and $w$ be independent probe vectors, and define $g = \tilde{K}^{-1}z$ and $h = \tilde{K}^{-1}w$. Then

$$\frac{\partial^2}{\partial\theta_i\partial\theta_j}\left[\log|\tilde{K}|\right] = \mathbb{E}\left[g^T\frac{\partial^2\tilde{K}}{\partial\theta_i\partial\theta_j}z - \left(g^T\frac{\partial\tilde{K}}{\partial\theta_i}w\right)\left(h^T\frac{\partial\tilde{K}}{\partial\theta_j}z\right)\right],$$

$$\frac{\partial^2}{\partial\theta_i\partial\theta_j}\left[(y - \mu_X)^T\alpha\right] = 2\mathbb{E}\left[\left(z^T\frac{\partial\tilde{K}}{\partial\theta_i}\alpha\right)\left(g^T\frac{\partial\tilde{K}}{\partial\theta_j}\alpha\right)\right] - \alpha^T\frac{\partial^2\tilde{K}}{\partial\theta_i\partial\theta_j}\alpha.$$

Hence, if we use the stochastic Lanczos method to compute the log determinant and its derivatives, the additional work required to obtain a second derivative estimate is one MVM by each second partial of the kernel for each probe vector and for $\alpha$, one MVM of each first partial of the kernel with $\alpha$, and a few dot products.

## 3.5 Radial basis functions

Another way to deal with the log determinant and its derivatives is to evaluate the log determinant term at a few systematically chosen points in the space of hyperparameters and fit an interpolation approximation to these values. This is particularly useful when the kernel depends on a modest number of hyperparameters (e.g., half a dozen), and thus the number of points we need to precompute is relatively small. We refer to this method as a surrogate, since it provides an inexpensive substitute for the log determinant and its derivatives. For our surrogate approach, we use radial basis function (RBF) interpolation with a cubic kernel and a linear tail. See e.g. [35–38] and the supplementary material for more details on RBF interpolation.

# 4 Error properties

In addition to the usual errors from sources such as solver termination criteria and floating point arithmetic, our approach to kernel learning involves several additional sources of error: we approximate the true kernel with one that enables fast MVMs, we approximate traces using stochastic estimation, and we approximate the actions of $\log(\tilde{K})$ and $\tilde{K}^{-1}$ on probe vectors.

We can compute first-order estimates of the sensitivity of the log likelihood to perturbations in the kernel using the same stochastic estimators we use for the derivatives with respect to hyperparameters. For example, if $\mathcal{L}^{\mathrm{ref}}$ is the likelihood for a reference kernel $\tilde{K}^{\mathrm{ref}} = \tilde{K} + E$, then

$$\mathcal{L}^{\mathrm{ref}}(\theta|y) = \mathcal{L}(\theta|y) - \frac{1}{2}\left(\mathbb{E}\left[g^T E z\right] - \alpha^T E \alpha\right) + O(\|E\|^2),$$

and we can bound the change in likelihood at first order by $\|E\|\left(\|g\|\|z\| + \|\alpha\|^2\right)$. Given bounds on the norms of $\partial E/\partial \theta_i$, we can similarly estimate changes in the gradient of the likelihood, allowing us to bound how the marginal likelihood hyperparameter estimates depend on kernel approximations.

If $\tilde{K} = U\Lambda U^T + \sigma^2 I$, the Hutchinson trace estimator has known variance [39]

$$\mathrm{Var}[z^T \log(\tilde{K})z] = \sum_{i\neq j}[\log(\tilde{K})]_{ij}^2 \leq \sum_{i=1}^n \log(1 + \lambda_j/\sigma^2)^2.$$

If the eigenvalues of the kernel matrix without noise decay rapidly enough compared to $\sigma$, the variance will be small compared to the magnitude of $\mathrm{tr}(\log \tilde{K}) = 2n\log\sigma + \sum_{i=1}^n \log(1 + \lambda_j/\sigma^2)$. Hence, we need fewer probe vectors to obtain reasonable accuracy than one would expect from bounds that are blind to the matrix structure. In our experiments, we typically only use 5–10 probes — and we use the sample variance across these probes to estimate *a posteriori* the stochastic component of the error in the log likelihood computation. If we are willing to estimate the Hessian of the log likelihood, we can increase rates of convergence for finding kernel hyperparameters.

The Chebyshev approximation scheme requires $O(\sqrt{\kappa}\log(\kappa/\epsilon))$ steps to obtain an $O(\epsilon)$ approximation error in computing $z^T \log(\tilde{K})z$, where $\kappa = \lambda_{\max}/\lambda_{\min}$ is the condition number of $\tilde{K}$ [19]. This behavior is independent of the distribution of eigenvalues within the interval $[\lambda_{\min}, \lambda_{\max}]$, and is close to optimal when eigenvalues are spread quasi-uniformly across the interval. Nonetheless, when the condition number is large, convergence may be quite slow. The Lanczos approach converges at least twice as fast as Chebyshev in general [20, Remark 1], and converges much more rapidly when the eigenvalues are *not* uniform within the interval, as is the case with log determinants of many kernel matrices. Hence, we recommend the Lanczos approach over the Chebyshev approach in general. In all of our experiments, the error associated with approximating $z^T \log(\tilde{K})z$ by Lanczos was dominated by other sources of error.

# 5   Experiments

We test our stochastic trace estimator with both Chebyshev and Lanczos approximation schemes on: (1) a sound time series with missing data, using a GP with an RBF kernel; (2) a three-dimensional space-time precipitation data set with over half a million training points, using a GP with an RBF kernel; (3) a two-dimensional tree growth data set using a log-Gaussian Cox process model with an RBF kernel; (4) a three-dimensional space-time crime datasets with a log-Gaussian Cox model with Matérn 3/2 and spectral mixture kernels; and (5) a high-dimensional feature space using the deep kernel learning framework [24]. In the supplementary material we also include several additional experiments to illustrate particular aspects of our approach, including kernel hyperparameter recovery, diagonal corrections (Section 3.3), and surrogate methods (Section 3.5). Throughout we use the SKI method [13] of Eq. (2) for fast MVMs. We find that the Lanczos and surrogate methods are able to do kernel recovery and inference significantly faster and more accurately than competing methods.

## 5.1   Natural sound modeling

Here we consider the natural sound benchmark in [13], shown in Figure 1(a). Our goal is to recover contiguous missing regions in a waveform with $n = 59,306$ training points. We exploit Toeplitz structure in the $K_{UU}$ matrix of our SKI approximate kernel for accelerated MVMs.

The experiment in [13] only considered scalable inference and prediction, but not hyperparameter learning, since the scaled eigenvalue approach requires all the eigenvalues for an $m \times m$ Toeplitz matrix, which can be computationally prohibitive with cost $\mathcal{O}(m^2)$. However, evaluating the marginal likelihood on this training set is not an obstacle for Lanczos and Chebyshev since we can use fast MVMs with the SKI approximation at a cost of $\mathcal{O}(n + m \log m)$.

In Figure 1(b), we show how Lanczos, Chebyshev and surrogate approaches scale with the number of inducing points $m$ compared to the scaled eigenvalue method and FITC. We use 5 probe vectors and 25 iterations for Lanczos, both when building the surrogate and for hyperparameter learning with Lanczos. We also use 5 probe vectors for Chebyshev and 100 moments. Figure 1(b) shows the runtime of the hyperparameter learning phase for different numbers of inducing points $m$, where Lanczos and the surrogate are clearly more efficient than scaled eigenvalues and Chebyshev. For hyperparameter learning, FITC took several hours to run, compared to minutes for the alternatives; we therefore exclude FITC from Figure 1(b). Figure 1(c) shows the time to do inference on the 691 test points, while 1(d) shows the standardized mean absolute error (SMAE) on the same test points. As expected, Lanczos and surrogate make accurate predictions much faster than Chebyshev, scaled eigenvalues, and FITC. In short, Lanczos and the surrogate approach are much faster than alternatives for hyperparameter learning with a large number of inducing points and training points.

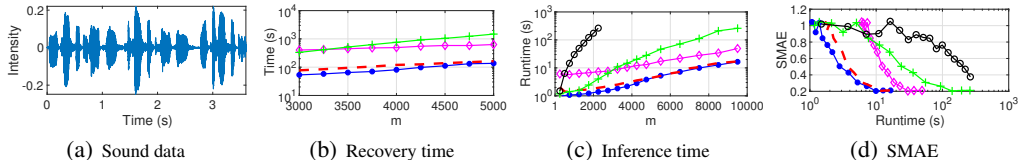

| (a) Sound data | (b) Recovery time | (c) Inference time | (d) SMAE |

Figure 1: Sound modeling using 59,306 training points and 691 test points. The intensity of the time series can be seen in (a). Train time for RBF kernel hyperparameters is in (b) and the time for inference is in (c). The standardized mean absolute error (SMAE) as a function of time for an evaluation of the marginal likelihood and all derivatives is shown in (d). Surrogate is (——), Lanczos is (- - -), Chebyshev is (— ⋄ —), scaled eigenvalues is (— + —), and FITC is (— o —).

## 5.2   Daily precipitation prediction

This experiment involves precipitation data from the year of 2010 collected from around $5500$ weather stations in the US[1]. The hourly precipitation data is preprocessed into daily data if full information of the day is available. The dataset has $628,474$ entries in terms of precipitation per day given the date, longitude and latitude. We randomly select $100,000$ data points as test points and use the remaining

points for training. We then perform hyperparameter learning and prediction with the RBF kernel, using Lanczos, scaled eigenvalues, and exact methods.

For Lanczos and scaled eigenvalues, we optimize the hyperparameters on the subset of data for January 2010, with an induced grid of 100 points per spatial dimension and 300 in the temporal dimension. Due to memory constraints we only use a subset of $12,000$ entries for training with the exact method. While scaled eigenvalues can perform well when fast eigendecompositions are possible, as in this experiment, Lanczos nonetheless still runs faster and with slightly lower MSE.

| Method | $n$ | $m$ | MSE | Time [min] |
|---|---|---|---|---|
| Lanczos | 528k | 3M | 0.613 | 14.3 |
| Scaled eigenvalues | 528k | 3M | 0.621 | 15.9 |
| Exact | 12k | - | 0.903 | 11.8 |

Table 1: Prediction comparison for the daily precipitation data showing the number of training points $n$, number of induced grid points $m$, the mean squared error, and the inference time.

Incidentally, we are able to use 3 *million* inducing points in Lanczos and scaled eigenvalues, which is enabled by the SKI representation [13] of covariance matrices, for a a very accurate approximation. This number of inducing points $m$ is unprecedented for typical alternatives which scale as $\mathcal{O}(m^3)$.

## 5.3 Hickory data

In this experiment, we apply Lanczos to the log-Gaussian Cox process model with a Laplace approximation for the posterior distribution. We use the RBF kernel and the Poisson likelihood in our model. The scaled eigenvalue method does not apply directly to non-Gaussian likelihoods; we thus applied the scaled eigenvalue method in [13] in conjunction with the Fiedler bound in [18] for the scaled eigenvalue comparison. Indeed, a key advantage of the Lanczos approach is that it can be applied whenever fast MVMs are available, which means no additional approximations such as the Fiedler bound are required for non-Gaussian likelihoods.

This dataset, which comes from the R package `spatstat`, is a point pattern of 703 hickory trees in a forest in Michigan. We discretize the area into a $60 \times 60$ grid and fit our model with exact, scaled eigenvalues, and Lanczos. We see in Table 2 that Lanczos recovers hyperparameters that are much closer to the exact values than the scaled eigenvalue approach. Figure 2 shows that the predictions by Lanczos are also indistinguishable from the exact computation.

| Method | $s_f$ | $\ell_1$ | $\ell_2$ | $-\log p(y|\theta)$ | Time [s] |
|---|---|---|---|---|---|
| Exact | 0.696 | 0.063 | 0.085 | 1827.56 | 465.9 |
| Lanczos | 0.693 | 0.066 | 0.096 | 1828.07 | 21.4 |
| Scaled eigenvalues | 0.543 | 0.237 | 0.112 | 1851.69 | 2.5 |

Table 2: Hyperparameters recovered on the Hickory dataset.

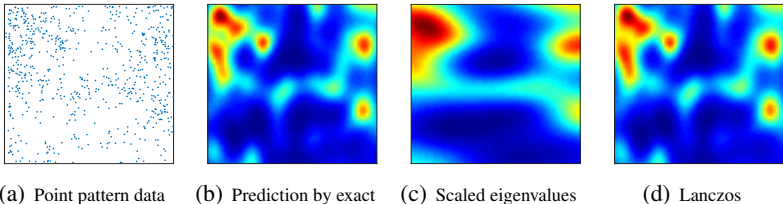

(a) Point pattern data   (b) Prediction by exact   (c) Scaled eigenvalues   (d) Lanczos

Figure 2: Predictions by exact, scaled eigenvalues, and Lanczos on the Hickory dataset.

## 5.4 Crime prediction

In this experiment, we apply Lanczos with the spectral mixture kernel to the crime forecasting problem considered in [18]. This dataset consists of $233,088$ incidents of assault in Chicago from January 1, 2004 to December 31, 2013. We use the first 8 years for training and attempt to predict the crime rate for the last 2 years. For the spatial dimensions, we use the log-Gaussian Cox process model, with the Matérn-5/2 kernel, the negative binomial likelihood, and the Laplace approximation for the

posterior. We use a spectral mixture kernel with 20 components and an extra constant component for the temporal dimension. We discretize the data into a $17 \times 26$ spatial grid corresponding to 1-by-1 mile grid cells. In the temporal dimension we sum our data by weeks for a total of 522 weeks. After removing the cells that are outside Chicago, we have a total of $157,644$ observations.

The results for Lanczos and scaled eigenvalues (in conjunction with the Fiedler bound due to the non-Gaussian likelihood) can be seen in Table 3. The Lanczos method used 5 Hutchinson probe vectors and 30 Lanczos steps. For both methods we allow 100 iterations of LBFGS to recover hyperparameters and we often observe early convergence. While the RMSE for Lanczos and scaled eigenvalues happen to be close on this example, the recovered hyperparameters using scaled eigenvalues are very different than for Lanczos. For example, the scaled eigenvalue method learns a much larger $\sigma^2$ than Lanczos, indicating model misspecification. In general, as the data become increasingly non-Gaussian the Fiedler bound (used for fast scaled eigenvalues on non-Gaussian likelihoods) will become increasingly misspecified, while Lanczos will be unaffected.

| Method | $\ell_1$ | $\ell_2$ | $\sigma^2$ | $T_{recovery}[s]$ | $T_{prediction}[s]$ | $RMSE_{train}$ | $RMSE_{test}$ |
|---|---|---|---|---|---|---|---|
| Lanczos | 0.65 | 0.67 | 69.72 | 264 | 10.30 | 1.17 | 1.33 |
| Scaled eigenvalues | 0.32 | 0.10 | 191.17 | 67 | 3.75 | 1.19 | 1.36 |

Table 3: Hyperparameters recovered, recovery time and RMSE for Lanczos and scaled eigenvalues on the Chicago assault data. Here $\ell_1$ and $\ell_2$ are the length scales in spatial dimensions and $\sigma^2$ is the noise level. $T_{recovery}$ is the time for recovering hyperparameters. $T_{prediction}$ is the time for prediction at all $157,644$ observations (including training and testing).

## 5.5 Deep kernel learning

To handle high-dimensional datasets, we bring our methods into the deep kernel learning framework [24] by replacing the final layer of a pre-trained deep neural network (DNN) with a GP. This experiment uses the gas sensor dataset from the UCI machine learning repository. It has 2565 instances with 128 dimensions. We pre-train a DNN, then attach a Gaussian process with RBF kernels to the two-dimensional output of the second-to-last layer. We then further train all parameters of the resulting kernel, *including* the weights of the DNN, through the GP marginal likelihood. In this example, Lanczos and the scaled eigenvalue approach perform similarly well. Nonetheless, we see that Lanczos can effectively be used with SKI on a high dimensional problem to train hundreds of thousands of kernel parameters.

| Method | DNN | Lanczos | Scaled eigenvalues |
|---|---|---|---|
| RMSE | $0.1366 \pm 0.0387$ | $0.1053 \pm 0.0248$ | $0.1045 \pm 0.0228$ |
| Time [s] | 0.4438 | 2.0680 | 1.6320 |

Table 4: Prediction RMSE and per training iteration runtime.

## 6 Discussion

There are many cases in which fast MVMs can be achieved, but it is difficult or impossible to efficiently compute a log determinant. We have developed a framework for scalable and accurate estimates of a log determinant and its derivatives relying only on MVMs. We particularly consider scalable kernel learning, showing the promise of stochastic Lanczos estimation combined with a pre-computed surrogate model. We have shown the scalability and flexibility of our approach through experiments with kernel learning for several real-world data sets using both Gaussian and non-Gaussian likelihoods, and highly parametrized deep kernels.

Iterative MVM approaches have great promise for future exploration. We have only begun to explore their significant generality. In addition to log determinants, the methods presented here could be adapted to fast posterior sampling, diagonal estimation, matrix square roots, and many other standard operations. The proposed methods only depend on fast MVMs—and the structure necessary for fast MVMs often exists, or can be readily created. We have here made use of SKI [13] to create such structure. But other approaches, such as stochastic variational methods [10], could be used or combined with SKI for fast MVMs, as in [26]. Moreover, iterative MVM methods naturally harmonize with GPU acceleration, and are therefore likely to increase in their future applicability and popularity. Finally, one could explore the ideas presented here for scalable higher order derivatives, making use of Hessian methods for greater convergence rates.

## Footnotes

[1]`https://catalog.data.gov/dataset/u-s-hourly-precipitation-data`

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
