[Supplementary Material · logdet-supp.pdf]

# Supplementary material: Scalable Log Determinants for Gaussian Process Kernel Learning

**Kun Dong**[1] **David Eriksson**[1] **Hannes Nickisch**[2] **David Bindel**[1] **Andrew Gordon Wilson**[1]
[1]Cornell University, [2]Phillips Research Hamburg

## 1 Background

Two popular covariance kernels are the RBF kernel

$$k_{\text{RBF}}(x, x') = s_f^2 \exp\left(\frac{\|x - x'\|^2}{2\ell^2}\right)$$

and the Matérn kernel

$$k_{\text{Mat},\nu}(x, x') = s_f^2 \frac{2^{1-\nu}}{\Gamma(\nu)} \left(\sqrt{2\nu}\frac{\|x-x'\|}{\ell}\right)^\nu K_\nu\left(\sqrt{2\nu}\frac{\|x-x'\|}{\ell}\right)$$

where $1/2$, $3/2$, and $5/2$ are popular choices for $\nu$ to model heavy-tailed correlations between function values. The spectral behavior of these and other kernels has been well-studied for years, and we recommend [5] for recent results. Particularly relevant to our discussion is a theorem due to Weyl, which says that if a symmetric kernel has $\nu$ continuous derivatives, then the eigenvalues of the associated integral operator decay like $|\lambda_n| = o(n^{-\nu-1/2})$. Hence, the eigenvalues of kernel matrices for the smooth RBF kernel (and of any given covariance matrix based on that kernel) tend to decay much more rapidly than those of the less smooth Matérn kernel, which has two derivatives at zero for $\nu = 5/2$, one derivative at zero for $\nu = 3/2$, and no derivatives at zero for $\nu = 1/2$. This matters to the relative performance of Chebyshev and Lanczos approximations of the log determinant for large values of $s_f$ and small values of $\sigma$ on the exact and approximate RBF kernel.

## 2 Methods

### 2.1 Scaled eigenvalue method

The scaled eigenvalue method was introduced in [7] to estimate $\log |K_{XX} + \sigma^2 I|$, where $X$ consists of $n$ points. The eigenvalues $\{\lambda_i\}_{i=1}^n$ of $K_{XX}$ can be approximated using the $n$ largest eigenvalues of a covariance matrix $\tilde{K}_{YY}$ on a full grid with $m$ points such that $X \subset Y$. Specifically,

$$\log |K_{XX} + \sigma^2 I| = \sum_{i=1}^n \log(\lambda_i + \sigma^2) \approx \sum_{i=1}^n \log\left(\frac{n}{m}\tilde{\lambda}_i + \sigma^2\right)$$

The induced kernel $K_{UU}$ plays the role of $\tilde{K}_{YY}$ when the scaled eigenvalue method is applied to SKI and the eigenvalues of $K_{UU}$ can be efficiently computed. Assuming that the eigenvalues can be computed efficiently is a much stronger assumption than our fast MVM based approach.

### 2.2 Radial basis function surrogates

Radial basis function (RBF) interpolation is one of the most popular approaches to approximating scattered data in a general number of dimensions [1, 2, 4, 6]. Given distinct interpolation points $\Theta = \{\theta^i\}_{i=1}^n$, the RBF model takes the form

$$s_\Theta(\theta) = \sum_{i=1}^n \lambda_i \varphi(\|x - \theta^i\|) + p(x) \tag{1}$$

where the kernel $\varphi : \mathbb{R}_{\geq 0} \to \mathbb{R}$ is a one-dimensional function and $p \in \Pi_{m-1}^d$, the space of polynomials with $d$ variables of degree no more than $m - 1$. There are many possible choices for $\varphi$ such as the cubic kernel $\varphi(r) = r^3$ and the thin-plate spline kernel $\varphi(r) = r^2 \log(r)$. The coefficients $\lambda_i$ are determined by imposing the interpolation conditions $s_\Theta(\theta^i) = \log|K(\theta^i)|$ for $i = 1, \dots, n$ and the discrete orthogonality condition

$$\sum_{i=1}^n \lambda_i q(\theta^i) = 0, \qquad \forall q \in \Pi_{m-1}^d. \tag{2}$$

For appropriate RBF kernels, this linear system is nonsingular provided that polynomials in $\Pi_{m-1}^d$ are uniquely determined by their values on the interpolation set.

## 2.3 Comparison to a reference kernel

Suppose more generally that $\tilde{K} = K + \sigma^2 I$ is an approximation to a reference kernel matrix $\tilde{K}^{\mathrm{ref}} = K^{\mathrm{ref}} + \sigma^2 I$, and let $E = K^{\mathrm{ref}} - K$. Let $\mathcal{L}(\theta|y)$ and $\mathcal{L}^{\mathrm{ref}}(\theta|y)$ be the log likelihood functions for the two kernels; then

$$\mathcal{L}^{\mathrm{ref}}(\theta|y) = \mathcal{L}(\theta|y) - \frac{1}{2}\left[\mathrm{tr}(\tilde{K}^{-1}E) - \alpha^T E \alpha\right] + O(\|E\|^2)$$

$$\frac{\partial}{\partial \theta_i}\mathcal{L}^{\mathrm{ref}}(\theta|y) = \frac{\partial}{\partial \theta_i}\mathcal{L}(\theta|y) - \frac{1}{2}\left[\mathrm{tr}\left(\tilde{K}^{-1}\frac{\partial E}{\partial \theta_i} - \tilde{K}^{-1}\frac{\partial \tilde{K}}{\partial \theta_i}\tilde{K}^{-1}E\right) - \alpha^T \frac{\partial E}{\partial \theta_i}\alpha\right] + O(\|E\|^2).$$

If we are willing to pay the price of a few MVMs with $E$, we can use these expressions to improve our maximum likelihood estimate. Let $z$ and $w$ be independent probe vectors with $g = \tilde{K}^{-1}z$ and $\hat{g} = \tilde{K}^{-1}w$. To estimate the trace in the derivative computation, we use the standard stochastic trace estimation approach together with the observation that $\mathbb{E}[ww^T] = I$:

$$\mathrm{tr}\left(\tilde{K}^{-1}\frac{\partial E}{\partial \theta_i} - \tilde{K}^{-1}\frac{\partial \tilde{K}}{\partial \theta_i}\tilde{K}^{-1}E\right) = \mathbb{E}\left[g^T \frac{\partial E}{\partial \theta_i}z - g^T \frac{\partial K}{\partial \theta_i}w\hat{g}^T E z\right]$$

This linearization may be used directly (with a stochastic estimator); alternately, if we have an estimates for $\|E\|$ and $\|\partial E/\partial \theta_i\|$, we can substitute these in order to get estimated bounds on the magnitude of the derivatives. Coupled with a similar estimator for the Hessian of the likelihood function (described in the supplementary materials), we can use this method to compute the maximum likelihood parameters for the fast kernel, then compute a correction $-H^{-1}\nabla_\theta \mathcal{L}^{\mathrm{ref}}$ to estimate the maximum likelihood parameters of the reference kernel.

# 3 Additional experiments

This section contains several experiments with synthetic data sets to illustrate particular aspects of the method.

## 3.1 1D cross-section plots

In this experiment we compare the accuracy of Lanczos and Chebyshev for 1-dimensional perturbations of a set of true hyper-parameters, and demonstrate how critical it is to use diagonal replacement for some approximate kernels. We choose the true hyper-parameters to be $(\ell, s_f, \sigma) = (0.1, 1, 0.1)$ and consider two different types of datasets. The first dataset consists of 1000 equally spaced points in the interval $[0, 4]$ in which case the kernel matrix of a stationary kernel is Toeplitz and we can make use of fast matrix-vector multiplication. The second dataset consists of 1000 data points drawn independently from a $U(0, 4)$ distribution. We use SKI with cubic interpolation to construct an approximate kernel based on 1000 equally spaced points. The function values are drawn from a GP with the true hyper-parameters, for both the true and approximate kernel. We use 250 iterations for Lanczos and 250 Chebyshev moments in order to assure convergence of both methods. The results for the first dataset with the RBF and Matérn kernels can be seen in Figure 1(a)-1(d). The results for the second dataset with the SKI kernel can be seen in Figure 2(a)-2(d).

Figure 1: 1-dimensional perturbations for the exact RBF and Matérn 1/2 kernel where the data is 1000 equally spaced points in the interval $[0, 4]$. The exact values are ($\bullet$), Lanczos is (——), Chebyshev is (——). The error bars of Lanczos and Chebyshev are 1 standard deviation and were computed from 10 runs with different probe vectors

Lanczos yields an excellent approximation to the log determinant and its derivatives for both the exact and the approximate kernels, while Chebyshev struggles with large values of $s_f$ and small values of $\sigma$ on the exact and approximate RBF kernel. This is expected since Chebyshev has issues with the singularity at zero while Lanczos has large quadrature weights close to zero to compensate for this singularity. The scaled eigenvalue method has issues with the approximate Matérn 1/2 kernel.

### 3.2 Why Lanczos is better than Chebyshev

In this experiment, we study the performance advantage of Lanczos over Chebyshev. Figure 3 shows that the Ritz values of Lanczos quickly converge to the spectrum of the RBF kernel thanks to the absence of interior eigenvalues. The Chebyshev approximation shows the expected equioscillation behavior. More importantly, the Chebyshev approximation for logarithms has its greatest error near zero where the majority of the eigenvalues are, and those also have the heaviest weight in the log determinant.

Another advantage of Lanczos is that it requires minimal knowledge of the spectrum, while Chebyshev needs the extremal eigenvalues for rescaling. In addition, with Lanczos we can get the derivatives with only one MVM per hyper-parameter, while Chebyshev requires an MVM at each iteration, leading to extra computation and memory usage.

### 3.3 The importance of diagonal correction

This experiment shows that diagonal correction of the approximate kernel can be very important. Diagonal correction cannot be used efficiently for some methods, such as the scaled eigenvalue method, and this may hurt its predictive performance. Our experiment is similar to [3]. We generate 1000 uniformly distributed points in the interval $[-10, 10]$, and we choose a small number of inducing points in such a way that there is a large chunk of the interval where there is no inducing point. We are interested in the behavior of the predictive uncertainties on this subinterval. The function values are given by $f(x) = 1 + x/2 + \sin(x)$ and normally distributed noise with standard deviation $0.05$ is

(a) log marginal likelihood for the RBF kernel      (b) log marginal likelihood for the Matérn kernel

(c) log determinant for the RBF kernel      (d) log determinant for the Matérn kernel

Figure 2: 1-dimensional perturbations with the SKI (cubic) approximations of the RBF and Matérn $1/2$ kernel where the data is 1000 points drawn from $\mathcal{N}(0, 2)$. The exact values are (•), Lanczos with diagonal replacement is (——), Chebyshev with diagonal replacement is (——), Lanczos without diagonal replacement is (——), Chebyshev without diagonal replacement is (——), and scaled eigenvalues is (×). Diagonal replacement makes no perceptual difference for the RBF kernel so the lines are overlapping in this case. The error bars of Lanczos and Chebyshev are 1 standard deviation and were computed from 10 runs with different probe vectors

added to the function values. We find the optimal hyper-parameters of the Matérn $3/2$ using the exact method and use these hyper-parameters to make predictions with Lanczos, Chebyshev, FITC, and the scaled eigenvalue method. We consider Lanczos both with and without diagonal correction in order to see how this affects the predictions. The results can be seen in Figure 4.

It is clear that Lanczos and Chebyshev are too confident in the predictive mean when diagonal correction is not used, while the predictive uncertainties agree well with FITC when diagonal correction is used. The scaled eigenvalue method cannot be used efficiently with diagonal correction and we see that this leads to predictions similar to Lanczos and Chebyshev without diagonal correction. The flexibility of being able to use diagonal correction with Lanczos and Chebyshev makes these approaches very appealing.

### 3.4 Surrogate log determinant approximation

The point of this experiment is to illustrate how accurate the level-curves of the surrogate model are compared to the level-curves of the true log determinant. We consider the RBF and the Matérn $3/2$ kernels and the same datasets that we considered in 3.1. We fix $s_f = 1$ and study how the level curves compare when we vary $\ell$ and $\sigma$. Building the surrogate with all three hyper-parameters produces similar results, but requires more design points. We use 50 design points to construct a cubic RBF with a linear tail. The values of the log determinant and its derivatives are computed with Lanczos. It is clear from Figure 5 that the surrogate model does a good job approximating the log determinant for both kernels.

### 3.5 Kernel hyper-parameter recovery

This experiments tests how well we can recover hyper-parameters from data generated from a GP. We compare Chebyshev, Lanczos, the surrogate, the scaled eigenvalue method, and FITC. We consider a

(a) True spectrum

(b) Lanczos weights

(c) Chebyshev weights

(d) Chebyshev absolute error

Figure 3: A comparison between the true spectrum, the Lanczos weights ($m = 50$), and the Chebyshev weights ($m = 100$) for the RBF kernel with $\ell = 0.3$, $s_f = 1$, and $\sigma = 0.1$. All weights and counts are on a log-scale so that they are easier to compare. Blue bars correspond to positive weights while red bars correspond to negative weights.

dataset of 5000 points generated from a $\mathcal{N}(0, 2)$ distribution. We use SKI with cubic interpolation and a total of 2000 inducing points for Lanczos, Chebyshev, and then scaled eigenvalue method. FITC was used with 750 equally spaced points because it has a longer runtime as a function of the number of inducing points. We consider the RBF kernel and the Matérn $3/2$ kernel and sample from a GP with ground truth parameters $(\ell, s_f, \sigma) = (0.01, 0.5, 0.05)$. The GPs for which we try to recover the hyper-parameters were generated from the original kernel. It is important to emphasize that there are two sources of errors present: the error from the kernel approximation errors and the stochastic error from Lanczos and Chebyshev. We saw in Figure 1 and 2 that the stochastic error for Lanczos is relatively small, so this follow-up experiment helps us understand how Lanczos is influenced by the error incurred from an approximate kernel. We show the true log marginal likelihood, the recovered hyper-parameters, and the run-time in Table 1.

It is clear from Table 1 that most methods are able to recover parameters close to the ground truth for the RBF kernel. The results are more interesting for the Matérn $3/2$ kernel where FITC struggles and the parameters recovered by FITC have a value of the log marginal likelihood that is much worse than the other methods.

(a) Lanczos with diagonal correction

(b) Lanczos without diagonal correction

(c) Chebyshev with diagonal correction

(d) Chebyshev without diagonal correction

(e) FITC

(f) Scaled eigenvalue method

Figure 4: Example that shows how important diagonal correction can be for some kernels. The Matérn $3/2$ kernel was used to fit the data given by the black dots. This data was generated from the function $f(x) = 1 + x/2 + \sin(x)$ to which we added normally distributed noise with standard deviation $0.05$. We used the exact method to find the optimal hyper-parameters and used these hyper-parameters to study the different behavior of the predictive uncertainties when the inducing points are given by the green crosses. The solid blue line is the predictive mean and the dotted red lines shows a confidence interval of two standard deviations.

(a) RBF exact

(b) Matérn 3/2 exact

(c) RBF surrogate

(d) Matérn 3/2 surrogate

Figure 5: Level curves of the exact and surrogate approximation of the log determinant as a function of $\ell$ and $\sigma$ for the RBF and Matérn 3/2 kernels. We used $s_f = 1$ and the dataset consisted of 1000 equally spaced points in the interval $[0, 4]$. The surrogate model was constructed from the points shown with ($\bullet$) and the log determinant values were computed using stochastic Lanczos.

| | | RBF | Matérn 3/2 |
|---|---|---|---|
| True | $-\log p(y\|\theta)$ | $-6.22\mathrm{e}3$ | $-4.91\mathrm{e}3$ |
| | Hypers | $(0.01, 0.5, 0.05)$ | $(0.01, 0.5, 0.05)$ |
| Exact | $-\log p(y\|\theta)$ | $-6.23\mathrm{e}3$ | $-4.91\mathrm{e}3$ |
| | Hypers | $(1.01\mathrm{e}{-}2, 4.81\mathrm{e}{-}1, 5.03\mathrm{e}{-}2)$ | $(9.63\mathrm{e}{-}3, 4.87\mathrm{e}{-}1, 4.96\mathrm{e}{-}2)$ |
| | Time (s) | $368.9$ | $466.7$ |
| Lanczos | $-\log p(y\|\theta)$ | $-6.22\mathrm{e}3$ | $-4.86\mathrm{e}3$ |
| | Hypers | $(1.00\mathrm{e}{-}2, 4.77\mathrm{e}{-}1, 5.03\mathrm{e}{-}2)$ | $(1.04\mathrm{e}{-}2, 4.87\mathrm{e}{-}1, 4.67\mathrm{e}{-}2)$ |
| | Time (s) | $66.2$ | $133.4$ |
| Chebyshev | $-\log p(y\|\theta)$ | $-6.23\mathrm{e}3$ | $-4.81\mathrm{e}3$ |
| | Hypers | $(9.84\mathrm{e}{-}3, 4.85\mathrm{e}{-}1, 5.12\mathrm{e}{-}2)$ | $(1.11\mathrm{e}{-}2, 4.66\mathrm{e}{-}1, 5.78\mathrm{e}{-}2)$ |
| | Time (s) | $110.3$ | $173.3$ |
| Surrogate | $-\log p(y\|\theta)$ | $-6.22\mathrm{e}3$ | $-4.86\mathrm{e}3$ |
| | Hypers | $(1.01\mathrm{e}{-}2, 4.88\mathrm{e}{-}1, 4.85\mathrm{e}{-}2)$ | $(1.02\mathrm{e}{-}2, 4.80\mathrm{e}{-}1, 4.66\mathrm{e}{-}2)$ |
| | Time (s) | $48.2$ | $44.3$ |
| Scaled eigenvalues | $-\log p(y\|\theta)$ | $-6.22\mathrm{e}3$ | $-4.71\mathrm{e}3$ |
| | Hypers | $(1.04\mathrm{e}{-}2, 4.52\mathrm{e}{-}1, 5.14\mathrm{e}{-}2)$ | $(1.13\mathrm{e}{-}2, 4.53\mathrm{e}{-}1, 6.37\mathrm{e}{-}2)$ |
| | Time (s) | $90.2$ | $127.3$ |
| FITC | $-\log p(y\|\theta)$ | $-6.22\mathrm{e}3$ | $-4.11\mathrm{e}3$ |
| | Hypers | $(1.03\mathrm{e}{-}2, 4.90\mathrm{e}{-}1, 5.07\mathrm{e}{-}2)$ | $(1.34\mathrm{e}{-}2, 5.22\mathrm{e}{-}1, 8.91\mathrm{e}{-}2)$ |
| | Time (s) | $86.6$ | $136.9$ |

Table 1: Hyper-parameter recovery for the RBF and Matérn 3/2 kernels. The data was generated from 5000 normally distributed points. Lanczos, surrogate, and scaled eigenvalues all used 2000 inducing points while FITC used 750. These numbers where chosen to make their run times close to equal. Diagonal correction was applied to the Matérn 3/2 approximate kernel. The value of the log marginal likelihood was was computed from the exact kernel and shows the value of the hyper-parameters recovered by each method. We ran Lanczos 5 times and averaged the values.