[Reviews · NeurIPS 2017]

Reviewer 1



Summary of the paper: This paper describes two techniques that can be used to estimate (stochastically) the log determinant of a positive definite matrix and its gradients. The aim is to speed up the inference process several probabilistic models that involve such computations. The example given is a Gaussian Process in which the marginal likelihood has to be maximized to find good hyper-parameters. The methods proposed are based on using the trace of the logarithm of a matrix to approximate the log determinant. Two techniques are proposed. The first one based on Chevyshev polynomials and the second one based on the Lanczos algorithm. The first one seems to be restricted to the case in which the eigenvalues lie in the interval [-1,1]. The proposed method are compared with sparse GP approximations based on the FITC approximation and with related methods based on scaled eigenvalues on several experiments involving GPs. Detailed comments: Clarity: The paper is very well written and all details seem to be clear with a few exceptions. For example, the reader may not be familiar with the logarithm of a matrix and similar advanced algebra operations. There are few typos in the paper also. In the Eq. above (2) it is not clear what beta_m or e_m are. They have not being defined. It is also known that the Lanczos algorithm is unstable and extra computations have to be done. The authors may comment on these too. Quality: I think the quality of the paper is high. In particular, the authors analyze the method described in several experiments, comparing with other important and related methods. These experiments seem to indicate that the proposed method gives competitive results at a small computational cost. Originality: The paper a strong related work section and the methods described seem to be original. Significance: My main concern with this paper is its significance. While it is true that the proposed method allow for a faster evaluation of the log determinant of the covariance matrix and its gradients. One still has to compute such a matrix with squared cost in the number of samples (or squared cost in the number of inducing points, if sparse approximations are used). This limits a bit the practical applicability of the proposed method. Furthermore, it seems that the experiments carried out by the authors involve a single repetition. Thus, there are no error bars in the results and one can not say whether or not the results are statistical significant. The differences with respect to the scaled eigenvalues problems (ref. [32]) in the paper seem very small.

Reviewer 2



This paper extends the Chebyshev method [9] and Lanczos method [28] for computing log-determinants, to hyperparameter learning that involves derivatives in gradient based optimization. The methods are not completely novel, but the experimental evaluations are solid and they reveal the superiority of Lanczos over Chebyshev and other compared methods, including scaled eigenvalues and FITC. I find that the paper is overall well presented. The authors may want to elaborate on the currently very brief section 3.4, which discusses the surrogate method. For example, "interpolation with a cubic basis function and a linear tail" is less than clear, and the method to "approximate derivatives of the log determinant using the surrogate" is also too obscure.

Reviewer 3



This is a well written paper that introduces a new method for approximating log determinants for GP hyperparameter estimation or model comparison when the number of observation points is large. It is based on Stochastic trace estimators (Monte Carlo) and Gauss quadratures. Furthermore, one asset of the method is to provide an estimate of the log determinants gradients for a reasonable extra cost. The effectiveness of the method is illustrated on several diverse examples where it is compared to the scaled eigenvalue method and exact computations when available. The question of speeding up the kernel parameter inference is of great importance to my opinion and it is sometimes overlooked when dealing with massive datasets. The proposed approach seems to be a sensible alternative to stochastic gradient methods. However, a comparison (or at least a list of pros and cons) between these two methods is kind of missing in the paper. The method is clearly presented and the illustrations on the examples are convincing. My only worry is about the method is the first step that consists in approximating the kernel by one allowing Matrix Vector Multiplication. This introduces a rather strong limitation to when the proposed method can be applied. Methods based on variational approximations of the posterior have become increasingly popular over the past years, would the proposed method still be applicable in this case?